# Anchor Loss Reduction of Lamb Wave Resonator by Pillar-Based Phononic Crystal

**DOI:** 10.3390/mi12010062

**Published:** 2021-01-07

**Authors:** Yinjie Tong, Tao Han

**Affiliations:** School of Electronic, Information and Electrical Engineering, Shanghai Jiao Tong University, Shanghai 200240, China; lomeijiave@sjtu.edu.cn

**Keywords:** anchor loss, MEMS resonator, phononic crystal, high quality factor

## Abstract

Energy leakage via anchors in substrate plates impairs the quality factor (*Q*) in microelectromechanical system (MEMS) resonators. Most phononic crystals (PnCs) require complicated fabrication conditions and have difficulty generating a narrow bandgap at high frequency. This paper demonstrates a pillar-based PnC slab with broad bandgaps in the ultra high frequency (UHF) range. Due to Bragg interference and local resonances, the proposed PnC structure creates notably wide bandgaps and shows great advantages in the high frequency, large electromechanical coupling coefficient ($k^{2}$) thin film aluminum nitride (AlN) lamb wave resonator (LWR). The dispersion relations and the transmission loss of the PnC structure are presented. To optimize the bandgap, the influence of the material mechanical properties, lattice type, pillar height and pillar radius are explored. These parameters are also available to adjust the center frequency of the bandgap to meet the desirable operating frequency. Resonators with uniform beam anchors and PnC slab anchors are characterized. The results illustrate that the *Q* of the resonator improves from 1551 to 2384, and the mechanical energy leakage via the anchors is significantly decreased using the proposed PnC slab anchors. Moreover, employment of the PNC slab anchors has little influence on resonant frequency and induces no spurious modes. Pillar-based PnCs are promising in suppressing the anchor loss and further improving the *Q* of the resonators.

## 1. Introduction

The fifth-generation of mobile networks (5G) has developed rapidly in recent years and bring new challenges in wireless communication and remote sensing applications [1,2,3]. This 5G technology has spurred the vast need for high-frequency wideband amplifiers and filters due to many appealing factors, including large $k^{2}$, complementary metal–oxide–semiconductor (CMOS)-compatibility and multi-frequency capability. The aluminum nitride (AlN) lamb wave resonator (LWR) is a competitive technology and has gained tremendous attention in the field of microelectromechanical system (MEMS) resonators. Among the various lamb wave modes propagating in the AlN plate, the lowest order symmetric (S0) mode and S1 mode stand out for their high phase velocity, weak velocity dispersion and excellent transduction efficiency [4]. Considering the utility in wireless communication, the membrane-based LWR in the 300–800 MHz range is highly desirable [2,5,6]. For LWRs, high quality factor (*Q*) is always considered to be a key factor for a large and diverse series of applications, including small insertion-loss bandgap filters and high resolution sensors.

Anchor loss has been reported to be the primary cause for the limitation of *Q* at high frequencies [7,8]. Anchor loss represents the leakage of a fraction of acoustic wave energy through the supporting anchors that hold the resonator body suspended, as shown in Figure 1a [7,8,9,10]. Approaches have been adopted in MEMS resonators to suppress the anchor loss to improve the *Q*. Pandey et al. designed a mesa surrounding the resonator to reflect the elastic energy back to the resonator [11]. Ardito et al. used holey phononic crystals (PnCs) to confine the wave energy to the resonator cavity [12]. Zou et al. demonstrated a butterfly-shaped resonator to diminish the vibration around the anchors [2]. Cassella et al. proposed a method by etching slots to limit the energy dissipation to the substrate [13]. Harrington et al. showed an in-plane acoustic wave reflector to form acoustic impedance discontinuity from the anchor to the substrate [14]. However, etching and pitting in piezoelectric material increase the manufacturing requirements. Some approaches cause spurious modes in the vicinity of the resonant frequency. Moreover, these methods present no obvious advantages for the high operational frequency LWRs. Further effective design is urgently needed to minimize the anchor loss to make a well-performing resonator based on the theoretical and experiment results [7,15].

The PnC anchor is a promising approach to reduce anchor loss, since it is able to form complete bandgaps in which the propagations of the acoustic waves are prohibited. However, most current PnCs require the membrane thickness to be approximately the same size as the lattice constant to achieve sufficient bandwidth [16,17,18]. Studies of PnC-based anchors have rarely been conducted when the scale of the membrane is 1 μm or even less. Moreover, most PnCs require stringent fabrication conditions at high frequencies [19]. Most importantly, the acoustic bandgap that is formed is not wide enough to cover the whole frequency domain between the resonant frequency and the anti-resonant frequency. Herein, the ideal design of PnCs for the further improvement of quality factors for thin film AlN LWRs is worth exploring.

In this work, a novel two-dimensional pillar-based PnC structure, acting as the anchor of an LWR to suppress anchor loss, is proposed. As shown in Figure 1b, a periodic array of scatterers composed of platinum (Pt) pillars is formed on top of the AlN substrates. The PnC slab structure provides insulation for acoustic waves and at the same time serves as the support anchor of the resonator. Our design (1) generates wide bandgaps in the ultra high frequency (UHF) range and provides great immunity to fabrication uncertainty, (2) mitigates restrictions on the film thickness, (3) has no sudden changes in geometry, (4) provides rather large freedoms to adjust the acoustic bandgaps and (5) eliminates the stress concentration effect usually existing in the structure with sharp corners. These advantages are particularly suitable for the high frequency, large $k^{2}$ thin film AlN LWR. Furthermore, thanks to the existence of gaps resulting from Bragg interference or from local resonances, the proposed PnC is favorable to form wide and multiple bandgaps and is able to provide multi-band applications and to expand the vibration reduction range. The PnC structures block the acoustic wave transmission via the anchors so that the anchor loss is well suppressed. The reduction of the insertion loss and the enhancement of the quality factor of the resonators are realized.

In Section 2, the design of the PnC anchors is described. The analysis of the dispersion relations and the transmission spectra are introduced and utilized. The influences of material mechanical properties, lattice types, pillar height and pillar radius are thoroughly discussed. In Section 3, the LWRs are designed, and the perfect match layer (PML)-based method to analyze anchor loss is introduced. Section 4 illustrates the finite element method (FEM) results with detailed discussion. Furthermore, the comparison between our results and the experimental results in the reference is presented in the supporting material [20]. Finally, Section 5 draws conclusions of our works.

## 2. Design of the Phononic Crystal Anchors

### 2.1. Dispersion Relations

PnCs are inhomogeneous structures where acoustic properties change periodically. It is well-known that well-designed PnC structures provide a salient phenomenon called bandgap where the acoustic wave propagation and mechanical vibration are forbidden [21,22,23,24]. Therefore, the utilization of the PnC slabs can efficiently attenuate the energy leaked via the anchors and restrict the mechanical vibration in the resonant body, which contributes to the improvement of the electrical performance of the resonator. The mechanical properties of the material, the geometrical size and the filling fraction all influence the existence and distribution of the phononic bandgap [25,26,27]. The pillar-based structure is essential to generate locally-resonant bandgaps by the coupling of flexural and torsional modes. The pillars serve as distributed springs and provide natural local resonators. Bragg interference and local resonators are two main mechanisms for opening bandgaps. Bragg interferences of waves are scattered at every periodic plane of scatterers (i.e., pillar) inside a crystal. When a forward and a backward propagating Bloch wave are phase matched but do not couple, they degenerate in the energy band structure space and generate Bragg bandgaps. Thus, Bragg bandgaps are open at frequencies directly related to the periodicity. Moreover, the local resonance of the scatterers can effective store and delay the incoming wave before the energy is radiated away to all permitted propagation modes. In this case, the positions of the bandgaps are conditioned by the resonance frequency of the individual cylindrical pillars. We investigated the propagation of an elastic wave in the PnC formed by a 1-μm-thickness Pt pillar on a thin AlN membrane and discovered that a square unit cell with lattice constant *a* = 4 μm creates wide bandgaps with a relatively small filling fraction. This pillar-based structure has the following advantages over the holey or fractal structure: (1) the generation of desired large complete bandgaps; (2) the guarantee of the structure strength for the remaining of the origin structure geometry and no etching or hollowing to undermine the structural integrity; (3) a further available design factor (pillar height) to adjust the bandgap; and (4) the possibility to create phononic bandgaps formed by both Bragg interference and local resonances simultaneously [24,28]. Figure 2a illustrates the basic unit and the detailed geometrical dimensions of the pillar-based PnCs. The piezoelectric membrane height e = 1 μm is fixed to meet the thickness of the resonator. The remaining geometry is designed to alleviate the fabrication requirement and in the meantime to provide wide complete acoustic bandgaps that agree strongly with the resonant frequency of the LWRs. The filling fraction (FF) is $\pi d^{2}/4a^{2}=0.44$.

The acoustic wave propagation in the solid is governed by
(1)$$\frac{\partial }{\partial x_{j}}\left(c_{ijkl}\frac{\partial u_{k}}{\partial x_{l}}\right)=\rho \frac{\partial^{2}u_{i}}{\partial t^{2}}\left(i,j,k,l=1,2,3\right),$$
where ρ is the density, $u_{i}$ correspond to the displacement components, *t* stands for the time, $c_{ijkl}$ represent the elastic constants and $x_{j}$, (*j* = 1, 2, 3) denote the *x*, *y* and *z* axis of coordinates, respectively. Moreover, the Bloch theorem is applied to analyze dispersive relations between the wave number *k* and the frequency *ω* to determine the dynamic characteristics of the structure. The Bloch wave propagating in a PnC can be expressed by the following form [29]:(2)$$u\left(r\right)=e^{ik\cdot r}\tilde{u}\left(r\right),$$
where *u* is the displacement field, *k* represents the Bloch wave vector and $\tilde{u}\left(r\right)$ stands for the Bloch displacement function with the same periodicity as the PnC. Due to the regular geometric characteristic and the high symmetry of the basic unit, the Brillouin zone is further reduced into the irreducible Brillouin zone. The analysis of the Bloch waves inside the irreducible Brillouin zone is sufficient to cover all Bloch waves of a periodic structure. Figure 2b depicts the irreducible Brillouin zone shaded in gray. The bandgaps are the frequency ranges inside which Bloch waves are absent and thus propagation is forbidden. A partial bandgap is valid for at least one direction in the Brillouin zone. A complete bandgap is valid in the whole Brillouin zone. Those highly symmetric points along the irreducible Brillouin zone are defined as Γ(0, 0), X($\frac{\pi }{a}$, 0) and M($\frac{\pi }{a}$,$\;\frac{\pi }{a})$. The dispersion relations between the wave number k and the frequency ω are obtained by sweeping the path Γ → X → M → Γ. Accordingly, the dispersion curves of the proposed pillar-based PnCs are calculated. The parameters in the FEA software, COMSOL ${\mathrm{Multiphysics}}^{®}$ (5.5, COMSOL, Inc, Burlington, MA, USA), are defined as the lattice constant a = 4 μm, the X component of wave vector Kx=if(k<1,k∗pi/a,if(k<2,pi/a,(3−k)∗pi/a)) and the Y component of wave number Ky=if(k<1,0,if(k<2,(k−1)∗pi/a,(3−k)∗pi/a)). The floquet periodicity containing *Kx* and *Ky* is applied to the boundary along the *x* and *y* axis to match the Bloch theorem. The parameter sweeping for *k* is selected from 0 to 3 in eigen-frequency solvers to go along the irreducible Brillouin zone and obtain the dispersion relations.

### 2.2. Optimization of Phononic Crystal

The material properties influence the bandgap of the structure. For a solid–solid PnC, the mass density and the Young’s modules are two main mechanical properties to adjust bandgaps. In accordance with the piezoelectric material of the resonator, the membrane material of the PnCs was chosen as AlN. Four different metals, including Al, Mo, Ag and Pt, were taken into consideration for the pillar material. The detailed parameters of the material are exhibited in Table 1. Al and AlN have similar mass density but sharply different Young’s modulus. While Mo and AlN have almost the same Young’s modulus, the mass density of Mo is about 3 times that of AlN. Ag was chosen because it has a similar mass density as Mo and also has a similar Young’s modulus as Al. We selected Pt for its large difference with AlN in both density and Young modulus. Figure 3a–d depict the dispersion curves of various pillar materials. Figure 3a exhibits a small bandgap between 492 MHz and 517 MHz. Figure 3b illustrates the bandgap ranging from 366 MHz to 465 MHz. Two complete bandgaps can be seen in Figure 3c,d, namely 295–360 and 542–617 MHz. The widest bandgap in Figure 3d stretches from 472 to 601 MHz, and the other bandgap covers the frequency domain of 243–316 MHz. In conclusion, the following rules were drawn for the design of pillar-based PnC structure: (1) the large difference in elastic properties of membrane and pillar can lead to the existence of a complete bandgap, (2) one or more bandgaps can be designed by different material and (3) a wide bandgap can be generated by the huge difference in elastic properties. Among all the bandgaps mentioned above, the PnC slab with Pt pillar generates the widest bandgap. Conventionally, the larger bandgap provides better performance in prohibiting the wave propagation [30]. Moreover, Pt is a CMOS compatible metal. Therefore, Pt was chosen for the cylinder pillars. It is desirable that the first and second bandgaps both cover the working frequency domain of LWRs.

Lattice types have strong influences on the acoustic bandgap [26,31,32]. Square lattice, hexagonal lattice and honeycomb lattice are frequently accepted as the shapes of the primitive cell. The three mentioned structures have the same lattice constant *a* = 4 μm, diameter of pillar *d* = 3 μm, height of pillar *h* = 1 μm and the pillar material Pt. To compare phononic bandgaps in three different lattice types, the first 20th dispersion curves were calculated. Figure 4a–c depict the schematics for the integration of the unit cell shaped in square, hexagon and honeycomb in a standard platform. Each shape presents great potentials in filling a regular plane. Figure 4d–f illustrate the irreducible part of the Brillouin zone and the phononic band structure of different lattice types. The widest bandgap of the honeycomb lattice exists in the 490–589 MHz. The widest bandgap of the hexagonal lattice locates at 485–595 MHz. Accordingly, square lattice provides superior properties compared to hexagonal lattice and honeycomb lattice in pillar-based PnC design. The wide complete acoustic bandgaps in UHF and very high frequency (VHF) ranges in Figure 4d can efficiently prevent elastic wave propagation and remarkably expand the vibration attenuation range.

The center frequency $f_{C}$ and the bandgap width *BW* of the bandgap are the two key parameters of acoustic bandgaps, because the former determines the operating frequency and the latter influences the utilization and the ability of PnCs. The $f_{C}$ and *BW* are defined as follows:(3)$$f_{C}=\frac{f_{U}+f_{L}}{2},BW=f_{U}-f_{L},$$
where $f_{U}$ and $f_{L}$ stand for the upper and lower frequency boundaries of the bandgap, respectively. PnC slabs with square lattice show $f_{C}\;$= 536 MHz and *BW* = 129 MHz.

Intriguingly, the pillar-based PnC is flexible to adjust acoustic bandgaps to cover the various working frequency domains of the resonators by changing pillar geometries. The pillar provides both the mass loading effect and the action of the natural resonator. Due to the mass loading effect, the scatter waves are in phase with the incident waves, and reflection is enhanced on the reflection side, while the scatter waves are in phase opposition with the incident field, and transmission tends to be cancelled on the transmission side [30]. Thickness-resonant, flexural and torsional modes of the pillar are attributed to the natural modes that are clamped at the interface with the substrate and free at the other end of the scatterer [33]. The mechanical energy is well confined inside the pillars, and the transmission through the PnC structure is strongly reduced when resonance is reached. The thickness-resonant modes are affected by the pillar height, while the flexural modes are affected by the pillar radius. The different colors in Figure 5a–c stand for the corresponding bandgap ranges. As shown in Figure 5a, the ranges of the first and second bandgaps can be tuned by the pillar height h. According to the mass loading effect, the center frequency $f_{C}$ of both the first and second bandgaps decreases with h. Figure 5d illustrates the relationships between pillar height and characteristics of the second bandgap. *BW* sharply rose when h changed from 0.6 to 1.0 μm and then declined slowly. Pillar height h was set to 1 μm to attain the largest bandgap. Moreover, we analyzed pillar radius’ impact on the bandgaps of PnCs. Figure 5b describes the relationship between the frequency domain of the first three bandgaps and pillar radius *r*. Figure 5e shows how $f_{C}$ and *BW* of the second bandgap vary with r and proves that *r* = 1.5 μm is the best choice in our PnC design to create the widest bandgap. By optimizing pillar height and pillar radius, the desired acoustic bandgaps can be tuned. Moreover, the influence of the membrane thickness was further analyzed. Figure 5c shows the tendency of the bandgaps as a function of the relative membrane thickness. As displayed in Figure 5f, *BW* of the second bandgap gradually decreased as the relative membrane was increased. The center frequency $f_{C}$ of the second bandgap rose quickly at first and became steadily eventually. These properties make the proposed pillar-based PnC slabs a viable choice for thin film LWRs, thin film bulk acoustic resonators (FBARs) and multiplexers [33].

### 2.3. Transmission Spectra

By calculating the transmission spectra of elastic waves, the existence and the ability of the acoustic bandgaps in the proposed PnC anchors were evaluated. Unlike the infinite array of scatters used for dispersion relations, the model used for transmission spectra analysis is a finite array of scatters, as shown in Figure 6a. The exciting and receiving port are set at the opposite sides of the 4 × 6 basic units of the PnCs. The PMLs are set to the both edges of the structure in the FEM model, with a function to reduce the reflection effect of the acoustic wave. The fixed displacement in the x, y and z directions is attached to the interface of the exciting port to provoke the different mechanical waves. The free boundary conditions are set to the upper and lower boundaries. The transmission loss is defined by the following equation:(4)$$S_{21}=10log_{10}\left(\frac{P2}{P1}\right)=10log_{10}\left(\frac{d_{\mathrm{receiving}}^{2}}{d_{\mathrm{source}}^{2}}\right)\left(dB\right),$$
where *P*1 and *P*2 are the time average power of the exciting port and the receiving port, respectively, and $d_{\mathrm{receiving}}$ and $d_{\mathrm{source}}$ denote the displacement in the exciting and receiving surface, respectively.

Figure 6c shows the displacement distribution of the structure when the exciting frequency is 700 MHz and locates outside any bandgap. The PnC slabs show ~−1 dB power attenuation in *x*, *y* and *z* directions. As a contrast, Figure 6d depicts the displacement distribution of the structure when the frequency is 550 MHz, falling into the second acoustic bandgap. The incident waves are almost completely reflected and the mechanical energy is well confined at the exciting port. The transmission spectra through the proposed PnCs is depicted in Figure 6b. Wide peak attenuations in three polarized sources strongly agree in the complete bandgaps.

## 3. Lamb Wave Resonator Design and Finite Element Analysis

In order to prove the effectiveness of the pillar-based PnC anchors on restraining the anchor loss, the AlN LWRs with uniform beam anchors and PnC slab anchors are designed and analyzed. Two kinds of resonators are both composited with 1 μm piezoelectric layer AlN and 100-nm-thick Pt electrodes. The only difference of the two resonators are the anchors. Detailed dimensions of LWRs are listed in Table 2.

The $Q_{\mathrm{anchor}}$ represents the ratio of the energy restored in the resonator body to the energy dissipated via the anchors per circle. The higher $Q_{\mathrm{anchor}}$ means that better anchor loss are suppressed. The $Q_{\mathrm{anchor}}$ of the resonator is defined as:(5)$$Q_{\mathrm{anchor}}=\left|\frac{Re\left(\omega \right)}{2Im\left(\omega \right)}\right|,$$
where ω denotes the complex eigen-frequency of the desired mode. For the purpose of precisely evaluating the anchor loss, the acoustic wave transmitting through the anchors should be absorbed before it is reflected back to the resonators. Absorbing boundary condition(ABC) and PML are two common technologies used in FEM simulation [8]. PML provides more flexibilities over ABC in geometric dimensions and boundary conditions and is applied to explore the anchor loss. As shown in Figure 7, semi-cylinders containing both the substrate plates and PMLs are used to assume the semi-infinite substrates. The radius of the substrate plates and PMLs in the FEM model are 2 wavelengths (*λ*) and 3*λ*, respectively.

## 4. Results and Discussion

We employed eigen-frequency in the models to analyze the anchor loss. The $Q_{\mathrm{anchor}}$ of the lamb wave resonators with uniform beam anchors and PnC slab anchors were 5176 and 123,741, respectively. The low $Q_{\mathrm{anchor}}$ of the LWR with the uniform beam anchors indicated that a large proportion of acoustic energy was dissipated through the anchors. The wide complete bandgaps of the pillar-based PnC slabs contributed to peak attenuation in wave propagation via the anchors, and significantly improved the $Q_{\mathrm{anchor}}$. The $Q_{\mathrm{anchor}}$ was enhanced about 24 times, which suggests that the anchor loss in the resonator was negligible. Moreover, the PnC slab tethers enabled a high $Q_{\mathrm{anchor}}$, which was much bigger than the $Q_{\mathrm{anchor}}$ reported in the existing schemes [2]. In addition, obvious displacement was noticed in the anchors in Figure 8a, which indicates that a part of the acoustic wave was dissipated. To obtain a more vivid view of the energy leakage, we normalized the displacement in the resonator and defined the maximum displacement $d_{\max}$ as 0 dB. The logarithmic value $l_{1}$ for the displacement $d_{1}$ is governed by:(6)$$l_{1}=10*log_{10}\left(\frac{d_{1}}{d_{\max}}\right)$$

The logarithmic displacements of the two resonators are shown in Figure 8c,d. With the benefit of the pillar-based PnC anchors, the displacements surrounding the anchors were significantly reduced, and the mechanical energy was well restrained in the resonant cavity.

Frequency domain analysis was carried out in the models. The material loss was estimated by applying an isotropic mechanical damping of 3 × 10^−4^ [5]. As illustrated in Figure 9, the resonant frequencies of the LWR with uniform beam anchors and PnC slab anchors were around 532.731 and 532.785 MHz, respectively, which clearly indicated that the PnC anchors had a very slight effect on the resonant frequency. By calculating the ratio of the resonant frequency and the −3dB bandwidth, the Q for the LWR with uniform beam anchors and PnC slab anchors were 1551 and 2384, respectively. The employment of PnC slab anchors contributed to a 54% Q improvement. Most importantly, no spurious modes were induced in the vicinity of resonant frequency by PnC slab anchors.

Due to the broad complete bandgap at 472–601 MHz, the LWR with PnC slab anchors efficiently blocked the acoustic energy dissipation to the substrate and demonstrated a 54% Q improvement over the identical LWR with the uniform beam tethers. The peak attenuation in vibration displacement around the anchors and the results of quality factor calculation further proved the enhancement of the performance of the LWR by our proposed PnC anchors. It is worth noting that the PnC slab anchors only showed a slight influence on the resonant frequency and induced no extra spurious modes. The numerical simulation indicated the large potential of the proposed pillar-based PnC slab anchors to improve Q. The anchor loss was significantly reduced, and then the low-impedance and high-quality-factor LWRS in high frequency were able to be achieved.

## 5. Conclusions

In summary, a novel design of pillar-based PnCs with the wide bandgaps in both UHF and VHF (i.e., 24% and 26% bandwidth) is proposed. The FEA method is introduced to obtain the dispersion relations and the transmission spectra of the two-dimensional PnCs. We analyze the influences of the material mechanical properties, lattice types, pillar height and pillar radius on the bandgaps to optimize our design. By comparison in the supporting material, our results and the experimental results in the references are perfectly matched [20]. By replacing the uniform beam anchors with our optimized PnC slab anchors, the mechanical vibrations in the substrate plates outside the resonator are remarkably reduced. Moreover, no spurious modes are induced in the working frequency domain, and the resonant frequency remains almost the same. The $Q_{\mathrm{anchor}}$ is 24 times higher than the normal one, and a 54% *Q* improvement over the LWR is achieved. All the results prove that the anchor loss is well suppressed and the *Q* of the LWR is significantly enhanced by the employment of the pillar-based PnC anchors.

## Figures and Tables

**Figure 1 micromachines-12-00062-f001:**
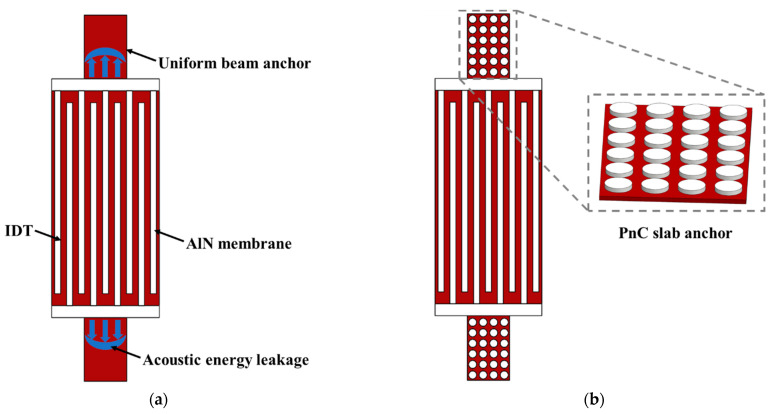
Illustrations of lamb wave resonators (LWRs) with (**a**) uniform beam anchors and (**b**) phononic crystal (PnC) slab anchors.

**Figure 2 micromachines-12-00062-f002:**
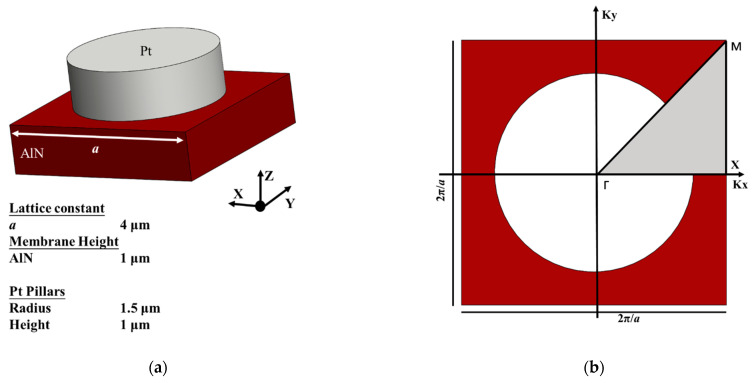
(**a**) Three-dimensional view of the basic unit of the PnC and the geometrical dimension. (**b**) The top view and the irreducible Brillouin zone of the basic unit.

**Figure 3 micromachines-12-00062-f003:**
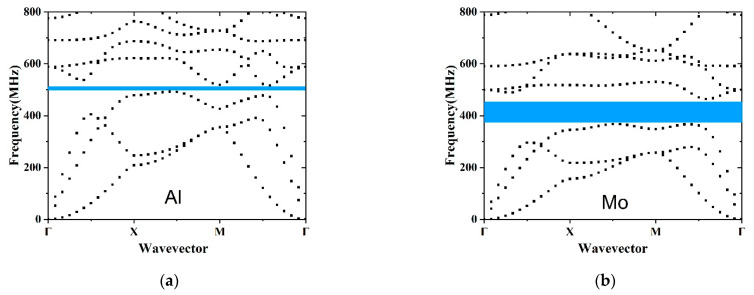
Dispersion relations of PnC slabs with (**a**) Al, (**b**) Mo, (**c**) Ag and (**d**) Pt pillars.

**Figure 4 micromachines-12-00062-f004:**
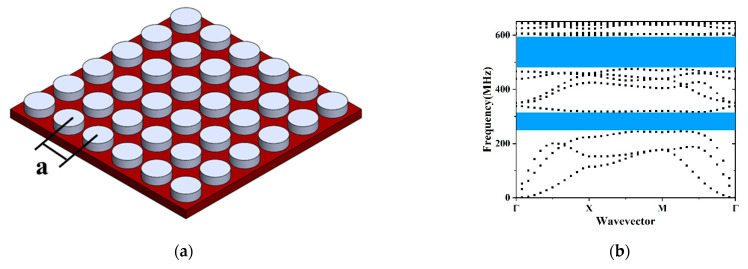
The schematics for (**a**) honeycomb, (**b**) hexagonal, (**c**) square lattice. The dispersion curves of PnC slabs with (**d**) honeycomb, (**e**) hexagonal, (**f**) square lattice.

**Figure 5 micromachines-12-00062-f005:**
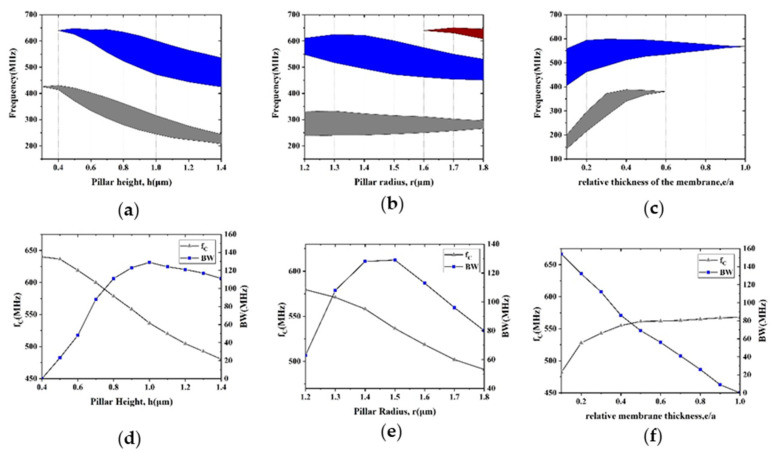
The dependence of the bandgap ranges on the (**a**) pillar height *h*, (**b**) pillar radius *r* and (**c**) relative membrane thickness. Characteristics of the second bandgap change with (**d**) pillar height *h* and (**e**) pillar radius *r* (**f**) relative membrane thickness.

**Figure 6 micromachines-12-00062-f006:**
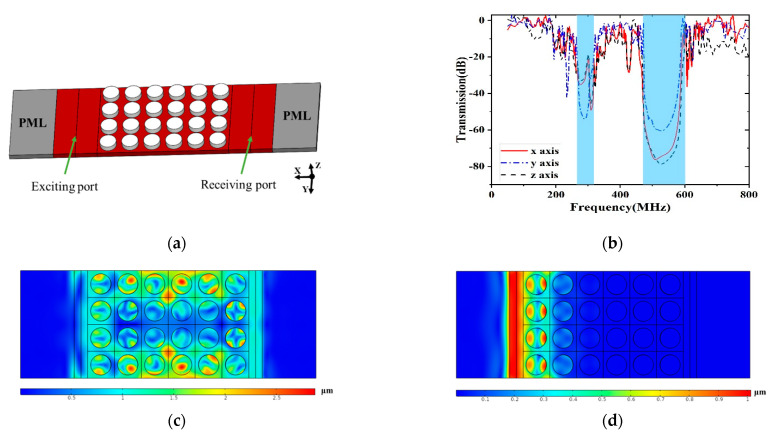
(**a**) Illustrations of the 4 × 6 pillar-based PnCs to analyze the transmission loss. (**b**) The simulated transmission spectra of acoustic wave via the PnC structure in *x*, *y* and *z* axes. (**c**) The displacement distribution at 700 MHz. (**d**) The displacement distribution at 550 MHz.

**Figure 7 micromachines-12-00062-f007:**
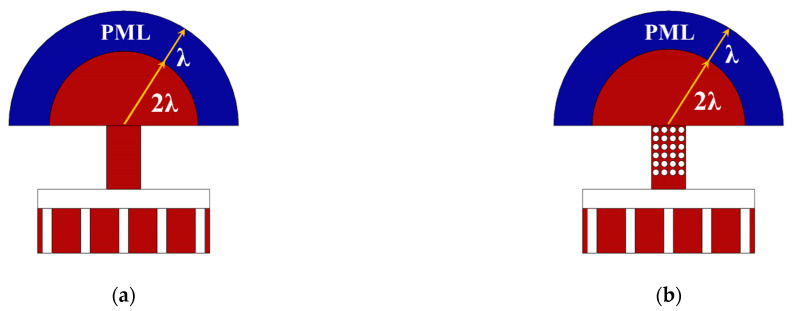
Illustrations of the lamb wave resonators with (**a**) uniform beam anchors and (**b**) PnC slab anchors attaching to the substrate plates covered by perfect match layers (PMLs).

**Figure 8 micromachines-12-00062-f008:**
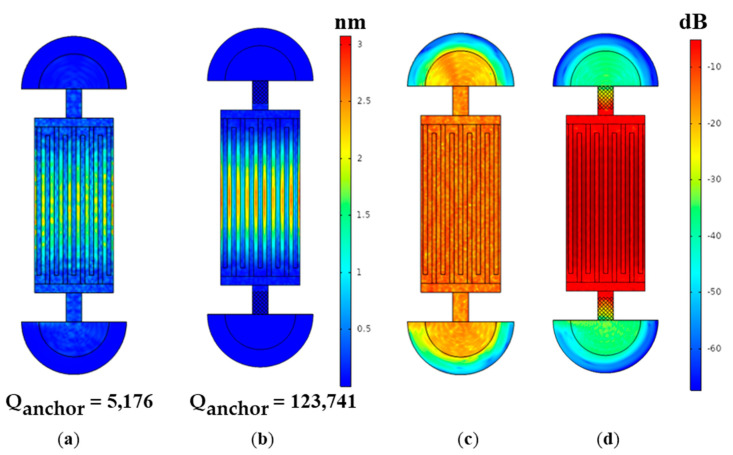
The displacement distribution of the LWR with (**a**) uniform beam anchors and (**b**) PnC slab anchors. The normalized logarithmic displacement distribution of the LWR with (**c**) uniform beam anchors and (**d**) PnC slab anchors.

**Figure 9 micromachines-12-00062-f009:**
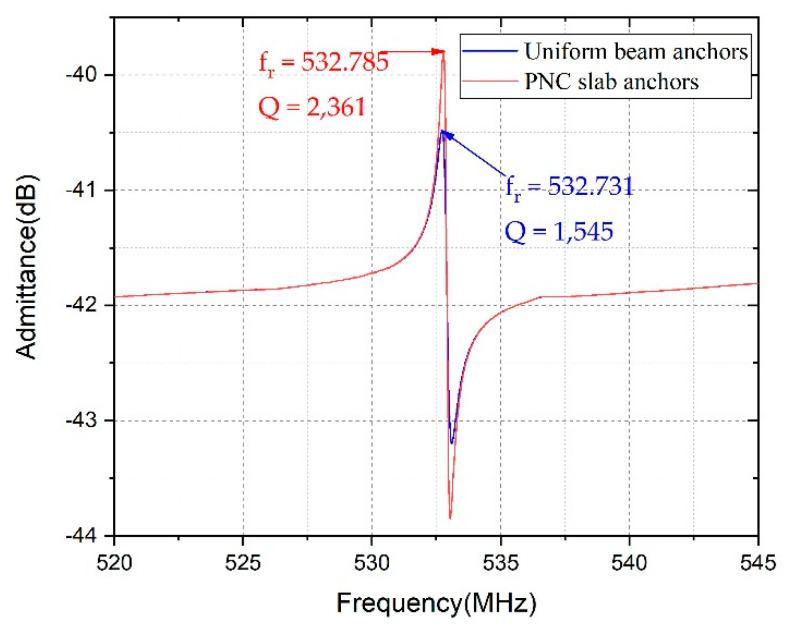
The admittance curves of the LWR with uniform beam anchors and PnC slab anchors.

**Table 1 micromachines-12-00062-t001:** Material characteristics.

Material	Mass Density (kg/m^3^)	Young’s Modulus (GPa)
Aluminum nitride (AlN)	3260	340
Aluminum (Al)	2700	70
Molybdenum (Mo)	10,200	330
Silver (Ag)	10,500	83
Platinum (Pt)	21,450	168

**Table 2 micromachines-12-00062-t002:** Parameters of the lamb wave resonator.

Parameters	Values
Inter digitated transducer (IDT) finger	9
Electrode pitch (μm)	9
Metal ratio	0.5
Aperture (μm)	180
Anchor length (μm)	30
Anchor width (μm)	16

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
