# Peer review of "Anchor Loss Reduction of Lamb Wave Resonator by Pillar-Based Phononic Crystal"

_micromachines, 2021, doi:10.3390/mi12010062_

Round 1
Reviewer 1 Report
I have read the review paper entitled “Anchor Loss Reduction of Lamb Wave Resonator by 2 Pillar-Based Phononic Crystal” by Tong Yinjie and Tao Han. This work tackles the issue of anchor losses from an interdigitated transducer system by proposing a phononic crystal based on regularly positioned metallic pillars. The authors explain that such structures can be very efficient while maintaining an ease of fabrication that other hole-based phononic crystals do not have. While the paper relies on the finite elements method to characterize and optimize their proposed design, the paper lacs comparison with the literature and does not clearly state in which way such phononic crystals are advantageous in terms of fabrication. The study itself is interesting but the paper lacks key elements, suffers from many approximations, and several sizeable mistakes are present in the text, therefore requiring substantial modifications.
Thus, I would recommend publication in Micromachines after major revisions.
General comments on the manuscript:
- The authors emphasize the advantage of pillar based PnCs over perforated PnCs for the anchors as it is supposedly easier to fabricate and requires less steps. However, they never detail a possible fabrication process for both cases to justify this claim. Whereas hole-based PnCs require a lithography and etching step, I would expect pillar-based PnCs to require a lithography step and a metal deposition step. This can be done at the same time as the transducers deposition only if the metal is the same. Could the authors clarify this?
- The authors are discussing a type of AlN Lamb wave resonator. In Line 30, they indicate advantages such as CMOS compatibility and multi-frequency ability. However, they never give details about the frequencies they aim at with their specific resonator, although it would determine which bandgap to choose.
The authors should therefore give more details at the beginning of their manuscript on what are the working frequencies of their LWR.
- There are several instances in the manuscript (see specific comments) in which the authors make claims that, although mostly reasonable, are not supported (in the manuscript) by values or references. Overall, the paper lacks quantitative analysis and comparison with existing schemes in terms of losses for instance.
Specific comments on the manuscript:
- The authors use bandgap, stop band and deaf band in the manuscript. Whereas deaf bands are a separate concept, it needs to be clarified earlier in the manuscript. Similarly, if authors mean different things by stop band and bandgap, they should clarify it. Otherwise, it will be clearer to stick to one appellation.
- L57: In the inset of Figure 1 I suggest increasing the pillar height and enhancing the perspective for clarity.
- L64-65: The authors make five statements in this sentence, but in particular the statement about fabrication uncertainty is not supported by data. If the authors mean that by slightly changing the dimensions of the pillars then the bandgap remains mostly unchanged, they should complete the text related to Figure 5 and quantify the sensitivity of the bandgap center frequency and bandwidth to the radius and height.
They also claim that it alleviates the restriction on the film thickness. This statement should be developed. Also, if film thickness becomes unrestricted, why don’t the authors use a-dimensional values.
- L78: Do the authors mean in the literature? If they refer to a specific reference, indicate which one.
- L91: Please indicate at least the pillar height in the text.
- L93: What fractal structures do the authors refer to?
- L97: The authors state that pillar-based PnCs have “the possibility to generate the bandgaps caused by both the Bragg interference and the local resonance simultaneously”. This is true and should be emphasized and developed slightly more in the manuscript.
- L108: Although well known, the authors should either detail the indices of the term cijkl as they do with the other terms, or write the equation in its general form.
- L118: The definition given by the authors for the bandgap is not accurate. Although it might be a typo mistake, please correct it.
- L123-124: The authors explain their mapping of the Brillouin zone by the k-vector. They write it in what I assume is the way they input it in the simulation software. The x and y components are written as having the exact same behavior over the whole Brillouin zone. This is certainly a mistake as kx and ky only have the same value on the MΓ Please correct this or include a small schematic to explain more clearly.
- The bandgap in Fig. 3a is not displayed at the correct position.
- L137: Are all the metals proposed CMOS compatible? Can the authors comment on this point when proposing the best material at the end of the manuscript?
- L148: It is probably due to a wrong use of the verb “attribute” but the sentence seems false to me. The bandgap is attributed to the difference in elastic properties, not the opposite as stated here.
- L152: A reference for larger bandgaps prohibiting wave propagation more efficiently would be useful.
- L170: The authors again emphasize the resistance to randomness in the distribution of the pillar. In what way does the dispersion relation “resist”? What is meant by broad randomness, a change from square to honeycomb and hexagonal lattice?
- L172-176: This should be defined much earlier in the text and also used for Figures 3 and 4.
- Figure 5: Panel a: What happens if you decrease the pillars height? Does the bandgap continue to increase in frequency? If so, wouldn’t it be something to point out? The authors could also briefly discuss the physical origin of the different bandgaps.
- L187: “r=1.5um is the best choice in our PnC design”. Best for what? The precise expectations of the LWR have not been defined in the paper. Please define what are the “optimal” conditions, such as expected working frequency of the LWR.
- Equation 4: d has not been defined in the text.
- Figure 6: All panels seem to have been compressed in one direction. Please resize the figure for proper viewing and reading of the titles.
“4×6 formed pillar-based PnC”: A better formulation can be found: such as removing “formed”.
- L204: What is a “sufficient inhibition”. This needs to be quantified and justified.
- Equation 5: The authors should add the absolute value to the fraction.
- L227-230: The description is fine but is Figure 7 really needed. A simplified schematic added to another figure would be enough.
- Figure 8: Is the whole part related to figure 8 here to validate the simulation performed in this work? If that is the case, this part should appear much earlier in the manuscript, before the results making use of these simulations. In this case, this figure is not needed and can be summarized in the text by a few sentences. The manuscript will benefit from less figures.
If not, the authors should clarify the purpose of this section.
Also, permission should be obtained to reproduce panels from the literature and the reference should be cited in the caption.
- L251: The authors state that the increase of 24 times of the anchor quality factor means that anchor losses in the resonator are negligible. It certainly means they are smaller, but what led to the conclusion that they are negligible. Can the authors quantify this?
The authors also state that a portion of acoustic waves is dissipated. What portion exactly?
- Figure 9: What is the difference between “displacement distribution” and “displacement level distribution”? In the first two panels, what does the m on top of the scale bar represent? The authors should include values for the scale bars.
- L260: How did the authors choose the isotropic mechanical damping value.
Typos:
In addition, there is a large number of typos in the text, some of the most important ones being indicated hereafter. I strongly suggest that the manuscript be edited by a native English speaker.
In the whole manuscript:
- the micrometer unit is written as um. This should be corrected to µm
- “Brillion” needs to be corrected into “Brillouin”
L11: “…PnC slab with wide bandgaps in the ultra high frequency…”
L19: “…illustrate that the Q…”
L27: “…(5G) has developed…”
L92: “unit” should be “unit cell”
L92: “performs” should be “generates” or “creates”
L96: “integrality” should be “integrity”
L115: “high symmetry” instead of “highly symmetry”
L215: The authors should precise that they are talking about the thicknesses.
Table 2: Why are “Anchor length” and “Anchor width” written in a bigger font than the rest of the table ?
Figure 9: Panels (c) and (d) are noted (a) and (b). Please correct.
Suggestions:
Two references that seem useful, to discuss Qfactor improvement and the physics of pillar based PnCs, respectively:
Muhammad Wajih Ullah Siddiqi and Joshua E.-Y. Lee, « Wide Acoustic Bandgap Solid Disk-Shaped Phononic Crystal Anchoring Boundaries for Enhancing Quality Factor in AlN-on-Si MEMS Resonators », Micromachines 9(8): 413 (2018).
Roman Anufriev & Masahiro Nomura « Phonon and heat transport control using pillar-based phononic crystals », Science and Technology of Advanced Materials, 19:1, 863-870 (2018).

Author Response
Dear Reviewer:
We would like to thank you for the valuable comments and suggestions on our manuscript. The authors express the deepest appreciation for the careful review and have carefully revised the paper based on the comments. The responses to comments and suggestions are in the attachment.
Thank you very much for your consideration! We are looking forward to hearing from you.
Yours sincerely,
Yinjie Tong,
on behalf of all authors

Reviewer 2 Report
The authors simulate the effect of using a phononic crystal (PC) to reduce anchor loss for AlN lamb wave resonators. Contrary to other anchor loss reduction with PC articles, they chose a pillar PC mainly for fabrication reasons and broad band gap. They simulate the effect of the material, height and radius of the pillars on the band gap. They also simulate the gain in the quality factor.
The author claim (line 64) that their design “provide great immunity to fabrication uncertainty”. But I saw no proof of that claim in this article. Maybe it is linked with ”the dispersion relation resists broad randomness in the distribution of the pillars” (line 170). However, they only show that different lattices lead to band gaps in the same frequency range, which is not randomness. They could simulate the effect of a non-homogeneity and a random dispersion in the height of the pillars. They could also simulate the effect of having non-straight pillar (base larger than the top for instance) which is classical when fabricating them.
The PC made of pillars to have the combination of both Bragg scattering mechanisms and local resonances are not new. A state of the art of these PC would have been a good idea.
It is Brillouin not Brillion!
Line 120: M coordinates are wrong.
Line 124: formula for Kx is the same as Ky and is therefore wrong. English should be use.
I do not see the point of lines 233 and following (fig. 8). What is not in the reference 25 is the transmission curve. What are the authors trying to prove? That they can make the same simulation of the dispersion curve? That a transmission spectrum will show band gaps (which is far from new)?
The authors mention a dead band in line 209. The choice of hatchings in fig. 6b make it difficult to see it. If they want to mention it, the authors should develop a bit more.
The authors should talk more about the mechanical damping (l. 260). Why this value?
Line 270 is confusing and may lead to think that fig. 10 is from experimental results (and since there is no MEB photo, I think this article is purely simulations)
Line 271 “transmission spectra analysis by our method” suggests that the authors developed a new method, which is not the case.
“d” should be define in equation (4)
It should be indicated that the upper and lower boundary (Y+ and Y-) conditions are free. The axis should be indicated in fig. 6.
Dimensions should be indicated in fig. 8a.
The number of the reference should be indicated in line 78 and 279
There are missing space, the use of um instead of µm for microns, point instead of coma (l. 30) or not well placed (l. 53) and some English mistakes.
In conclusion, I advise this article to be published after the aforementioned corrections.
Author Response

(The authors gave the same response as above.)

Reviewer 3 Report
This manuscript proposes using Pillar-based silicon photonic to improve the quality factor of MEMS AlN resonators. Through a comprehensive numerical simulation is it shown that a periodic array of Pt pillars on the top of the tethers can significantly reduce the anchor lost. The advantages of the pillar-based in comparison to other alternative approaches are highlighted. In general, the manuscript is written well, however, this work is the lack of prototype measurements. Consequently, advantages of the proposed method in the real environment are not clear.
- The introduction is quite short as well as weak. The authors have not addressed a wide range of works in this area. There are many papers in the literature that introduce the effect of anchor design on the Q-factor of a MEMS resonator. For instance, https://doi.org/10.1109/JMEMS.2009.2016271, https://doi.org/10.1109/ULTSYM.2016.7728780 and https://doi.org/10.1088/1361-6439/ab392c
- Figs.3 , Figs 4b, d and e, should be improved.
- 5a,b are vague. What do the different colors represent for? Does the author have any explanation for the relation of the pillar height and radius on the bandgap? Why the center frequency decreases with h? What is the physical explanation?
- It is not clear why/How the authors selected the mechanical damping as 3e-4? Where does this number come from?
- Since this study mainly focused on numerical simulation, COMSOL set up for each figure has to be clearly reported, in this way the potential readers can re-produce the results.
- 3 has no understandable caption. It is expected that the authors identify a, b, … in the caption.
- Although Fig. 3 demonstrates Bloch waves inside the irreducible Brillion zone, however, the authors have to clearly explain how these results obtained and what kind of software used.
- It is not clear why the authors only demonstrate the transmission along x axis in Fig. 8(b).
- 9 demonstrates the results graphically in m, and dB units. What dese it mean displacement in dB? Complementary information regarding displacement level distribution and displacement distribution is required.
- Eventually, the authors claimed that one of advantages of the proposed method is ease of fabrication. However, no information regarding the required micro-fabrication process is provided. Ideally, the authors have to manufacture the device and characterize it in the lab environment. In this regard, the FEM simulations will be verified and manufacturability of the proposed device can be clearly seen.
Author Response

(The authors gave the same response as above.)

Round 2
Reviewer 1 Report
Anchor Loss Reduction of Lamb Wave Resonator by Pillar-Based Phononic Crystal
Report:
I have read the revised version of the manuscript entitled “Anchor Loss Reduction of Lamb Wave Resonator by Pillar-Based Phononic Crystal” by Tong Yinjie and Tao Han. In its current form, the manuscript is of better quality than the first iteration. The authors have taken most of my comments into account. Although most points have been addressed, several of them were done in a too minimalistic fashion.
Most of my comments on the revised manuscript are follow up on previous questions and comments to which I would like the authors to enhance the completeness of their response.
Thus, I would recommend publication in Micromachines after minor revisions and a thorough check by a native English speaker.
General comment on the manuscript:
- The authors have clearly improved the general feel of the manuscript with a few sentences here and there, as well as references and a better choice of words in some instances. Nonetheless, although grammar mistakes and spelling have been improved, the level of English language strongly hampers the overall quality of this article and needs to be improved for ease of reading and understanding. Many sentences are understandable for someone in the field but will be obscure to non-specialists.
- Most of the answers provided to my comments are correct but lack depth. Oftentimes some words were changed in a sentence or one sentence added or removed. Whereas this is valid in several instances, there are other points for which a more thorough description is needed in the text. Specific comments are given below in this instance.
- In the discussion section, the authors should develop a bit more on the implications of their findings and potential improvements to this study. Overall the discussion is a bit light and the manuscript mostly presents the results.
Specific comments on the manuscript:
- The use of a-dimensional values in the manuscript is still basically non-existent. The authors have added the definition of one such value in the text but have not included it in Fig. 5 or in the manuscript beyond that point:
« Response 7: Thanks for your comment. The authors take the advice and use a-dimensional values. The contents are modified in Page2 and highlighted in the yellow,
“Our design 1) generates the wide bandgaps in UHF range and provide great immunity to fabrication uncertainty, 2) alleviates the restriction on the film thickness (i.e., e/a=0.25), 3) has no sudden changes in geometry, 4) provides rather large freedoms to adjust the acoustic bandgaps, 5) eliminates the stress concentration effect usually existing in the structure with sharp corners.” »
The authors need to provide a complete discussion on these parameters as well as potential scalability in terms of frequencies and sizes when the thickness of the membrane changes, which could be useful for other applications.
- In page 7, the authors have mentions of the effects of Bragg interferences and local resonances. This should be developed into a full paragraph, including the origin and the implications of these two effects that need to be discussed in more depth to fully highlight this specific advantage of pillar-based PnCs over holes. Similarly, mention can be added in the abstract and introduction to emphasize this point a bit more, although it is not specifically used in this work.
- In point 13 I had asked for a corrected definition of the bandgap. The new version remains inaccurate as the bandgap is a forbidden energy range between branches in the dispersion relation. This should appear in the manuscript. A quick discussion about complete and partial bandgaps would also be helpful in the manuscript. For instance, the authors could discuss the benefits of a complete bandgap in the framework of these LWRs.
- When discussing the origin of the bandgaps, the authors briefly mention the mass loading effect and the action of a natural resonator. Further description and references for these two effects would be useful to make the manuscript more complete.
As per one of there added references =, the authors could also discuss in the last section further improvements to their design, playing for instance on hybridized modes.
- Figure 6 is now more readable. However, the authors could add the value of displacement in panels c and d to give a clearer idea to the reader, as is done in Fig. 9.
- Following my point 24, the authors have rephrased the problematic expression. My meaning was that words such as “sufficient” or “obvious” do not have a well-defined meaning. To remedy that problem, the authors should give a quantitative value to the attenuation observed in their simulations.
- The figure in the manuscript are too disperse. For instance, Figs. 1 and 2 can be fused. In Figure 9, the horizontal space between panels a and b, and between c and d can be largely removed and the figure resized to fit in one row.
- 8 can come much earlier in the manuscript. I understand the importance of this section, as stated by the authors in their response but I have not understood why it should appear so late. Indeed, the authors simulate numerous things before that point and the simulation is then admitted to be accurate. I still believe advancing this section to an earlier point would show the reader the validity of the method before presenting and of the results that will then be accepted more readily by the readers.
Author Response
The authors express the deepest appreciation for the careful review and valuable comments. Thank you very much for your consideration! The responses to comments are in the attachment.

Reviewer 3 Report
The authors have addressed my previous comments, hence, the quality of the manuscript is improved and the reviewer has no more comments/concerns. However, this study is still a lack of experimental results, therefore, considering this paper for publishing is dependent on editor's decision.
Author Response
The authors express the deepest appreciation for the careful review and valuable comments. Thank you very much for your consideration!